# Compute-in-Memory for Numerical Computations

**DOI:** 10.3390/mi13050731

**Published:** 2022-05-02

**Authors:** Dongyan Zhao, Yubo Wang, Jin Shao, Yanning Chen, Zhiwang Guo, Cheng Pan, Guangzhi Dong, Min Zhou, Fengxia Wu, Wenhe Wang, Keji Zhou, Xiaoyong Xue

**Affiliations:** 1State Grid Key Laboratory of Power Industrial Chip Design and Analysis Technology, Beijing Smart-Chip Microelectronics Technology Co., Ltd., Beijing 100192, China; dongyan-zhao@sgitg.sgcc.com.cn (D.Z.); wangyubo@sgitg.sgcc.com.cn (Y.W.); shaojin@sgitg.sgcc.com.cn (J.S.); chenyanning@sgitg.sgcc.com.cn (Y.C.); pancheng@sgitg.sgcc.com.cn (C.P.); zhoumin3@sgitg.sgcc.com.cn (M.Z.); wangwenhe@sgitg.sgcc.com.cn (W.W.); 2Beijing Chip Identification Technology Co., Ltd., Beijing 100192, China; dongguangzhi@sgitg.sgcc.com.cn (G.D.); wufengxia@sgitg.sgcc.com.cn (F.W.); 3State Key Laboratory of ASIC and System, School of Microelectronics, Fudan University, Shanghai 201203, China; xuexiaoyong@fudan.edu.cn

**Keywords:** compute-in-memory (CIM), numerical computations, resistive random-access memory (ReRAM), partial differential equations (PDEs), crossbar

## Abstract

In recent years, compute-in-memory (CIM) has been extensively studied to improve the energy efficiency of computing by reducing data movement. At present, CIM is frequently used in data-intensive computing. Data-intensive computing applications, such as all kinds of neural networks (NNs) in machine learning (ML), are regarded as ‘soft’ computing tasks. The ‘soft’ computing tasks are computations that can tolerate low computing precision with little accuracy degradation. However, ‘hard’ tasks aimed at numerical computations require high-precision computing and are also accompanied by energy efficiency problems. Numerical computations exist in lots of applications, including partial differential equations (PDEs) and large-scale matrix multiplication. Therefore, it is necessary to study CIM for numerical computations. This article reviews the recent developments of CIM for numerical computations. The different kinds of numerical methods solving partial differential equations and the transformation of matrixes are deduced in detail. This paper also discusses the iterative computation of a large-scale matrix, which tremendously affects the efficiency of numerical computations. The working procedure of the ReRAM-based partial differential equation solver is emphatically introduced. Moreover, other PDEs solvers, and other research about CIM for numerical computations, are also summarized. Finally, prospects and the future of CIM for numerical computations with high accuracy are discussed.

## 1. Introduction

Different from data-intensive computing (typically all kinds of neural networks), high-precision computing is aimed at accurate numerical computations like large-scale matrix multiplication and partial differential equations (PDEs). At present, neural networks (NNs) have a lot of applications and are widely used in daily life. However, applications of high-precision computing could not be solved by NNs, whether in scientific research or in the actual scene. The requirements of throughput and energy efficiency for computing are constantly improving; therefore, CIM (computing-in-memory) is proposed as the solution of the Von Neumann bottleneck. The progress of CIM for numerical computations has great value in finance, engineering, computer science and other disciplines. It is ubiquitous in the field of scientific research and engineering. For example, improving the physical authenticity of virtual reality (VR), analyzing SIS infectious diseases with age structure, studying the BSM equations of derivative pricing theory, preprocessing and extracting image information and many other practical problems involve partial differential equations. In recent years, all kinds of PDEs solvers based on different CIMs, including ReRAM, SRAM, flash memory and PCM, have emerged in numerical computing research. ReRAM-based CIM, as a relatively mature CIM technology, is still used for most of the high-precision CIM research. So, this article reviews the ReRAM technology, the principle of the ReRAM crossbar and the working process of ReRAM in CIM, firstly. Then, it summarizes the numerical methods of PDEs, matrix iterative methods, rearrangement methods and split methods. After that, the working procedure and current developments of all kinds of CIM-based partial differential equation solvers are discussed. Moreover, their performance and characteristics are also compared. Aimed at defects in PDEs solvers, the solutions to get high-precision in large-scale matrix multiplication under environmental effects are proposed. In the future, the developments of CIM-based numerical computations will be improved in the manufacturing process, the write-verify method, the algorithm of sparse matrixes and the software/hardware collaboration. 

## 2. ReRAM

### 2.1. The Appearance of ReRAM

In the early 1960s, various research about ReRAM devices with all kinds of oxide materials, including Al_2_O_3_, NiO, SiO_2_, Ta_2_O_5_, ZrO_2_, TiO_2_ and Nb_2_O_5_, emerged in an endless stream [1,2,3,4,5]. Compared with metal-oxide-semiconductor field-effect transistors (MOSFET), which appeared in 1960 [6,7] for the first time, ReRAM devices are the products of the same period. In the 40 years that followed, the technology of resistive switching has not made significant progress in storage applications. 

### 2.2. The Development of ReRAM as NVM

With the explosive growth of portable electronic devices, the requirement and storage capacity of memory devices have increased rapidly. Higher density, faster speed and lower cost have become the goal of new memory devices. ReRAM, as a kind of nonvolatile memory (NVM) [8], was regarded as one of the continuations of NAND flash memory [9], though there were many emerging nonvolatile memory (eNVM) devices, such as phase-change memory (PCM) [10], magnetic random-access memory (MRAM) [11] and ferroelectric random-access memory (FeRAM) [12], over the same period. Table 1 lists the types of NVMs and the category of ReRAM. 

ReRAM is a two-terminal device with a variable resistance based on a physical mechanism of conducting filament formation and rupture [13]. According to the types of filamentary, ReRAM can be divided into oxide ReRAM (OxRAM) and conductive bridge ReRAM (CBRAM) [14]. ReRAM changes between high-resistance states (HRS) and low-resistance states (LRS) under different operating conditions, representing logic 0 and 1, separately. The formation of the conducting filament corresponds to the LRS, and the HRS is the opposite. 

Figure 1a is the structure of the OxRAM, and there is a metal oxide material between the two electrodes in the OxRAM. When a positive voltage is applied between the top electrode (TE) and the bottom electrode (BE), a conductive filament is formed between the two electrodes. While in Figure 1b, the electrode of the CBRAM is injected with copper or silver metal (Cu or Ag). Moreover, the CBRAM forms the conductive bridge by diffusing Cu or Ag into the oxide or chalcogenide (like GeS_2_). When the voltage is sufficiently positive, there will be the oxidation of Cu or Ag at TE, and they will be reduced and deposited at BE. When the voltage transfers to the negative, there will be the reduction of Cu or Ag at TE, and then the conductive filament connecting the two electrodes, and the state of the CBRAM, changes from HRS to LRS [15]. 

Since 2000, the research on ReRAM have explosively increased. The first NiO_x_-based ReRAM with promising device characteristics and reliability was proposed by I. Baek in 2005 [16]; the HfO_2_/Ti device was made with fully conventional fab materials [17]; the 3D vertical ReRAM emerged in 2009 [18]; the 10 × 10 nm^2^ Hf/HfO_x_ crossbar resistive RAM was produced in 2011 [19]; the first 16-Gb ReRAM integrated chip with copper oxide material [20], etc. However, because of the 15 nm critical dimension (CD) and the development of 3D NAND flash memory, using the ReRAM in high-density applications became more and more difficult.

### 2.3. ReRAM in CIM

In recent years, compute-in-memory widely emerged in machine learning (ML) and data-intensive computing. CIM is an effective method to break the Von Neumann bottleneck when computing large-scale data [21], which can achieve high speed and low-power computing by reducing data handling. Edge AI applications based on deep neural networks (DNNs) are designed to find the solution to achieving portable, fast, accurate and convenient computing. Computing efficiency (defined as terra-operations-per-second-per-millimeter-squared, TOPS/mm^2^) and energy efficiency (defined as terra-operations-per-second-per-watt, TOPS/W) are the two most significant parameters to measure the performance of computing. For digital neural network accelerators, multiplication and addition are calculated in the processing element (PE). However, the global buffer or cache is urgently needed to store the weights and the inputs/outputs, which increases the data storage and handling. There is quite a lot of research on optimizing data flow at the chip, micro and SOC (system on chip) levels, but computation and memory are all separated, which leads to efficiency degradation. The memory not only stores the weights and the inputs but also achieves analog computation. That is what computing-in-memory means. Compared with traditional digital signal accelerators, CIM as a mix-signal processing tremendously increases throughput, area efficiency and energy efficiency, but with the decline of accuracy. Though the requirement of analog-to-digital converters (ADCs) is inevitable, CIM still has enormous appeal in power consumption, no matter whether now or in the future.

Because of the resistive properties of ReRAM, ReRAM could be a natural electrical multiplier, with the function of storage following Ohm’s laws and Kirchhoff’s current laws (KCL). Utilizing I=V·G and the sum of the current, multiplication and accumulation are calculated by the ReRAM array in the analog domain, respectively. Obviously, eNVM, including resistive random-access memory, is an excellent memory device for CIM, and other eNVM devices such as PCM, MRAM and FeRAM are also of interest for CIM [22]. ReRAM is a better choice for compute-in-memory because of the 22 nm high reliability and the compatibility with the complementary metal-oxide-semiconductor (CMOS) process at present. In addition, ReRAM could potentially offer multi-bit per cell capability [22].

### 2.4. ReRAM Crossbar

Figure 2a,b shows the principle of calculation in the ReRAM crossbar array. One terminal of each RRAM is connected to the bit line (BL) collecting the current, and the other is connected to the word line (WL) as the input of the voltage. Additionally, the currents through the BLs are added as the outputs of the ReRAM array to achieve the accumulation. One of the outputs could be given by $I_{j}=\sum V_{i}\cdot G_{i,j}$ (where $I_{j}$ is the current of the jth column, $V_{i}$ is the input voltage of the ith row, and $G_{i,j}$ is the conductance of the ith row and jth column in ReRAM array). Due to the fact that the outputs are the analog currents, ADC is an indispensable part of the ReRAM crossbar peripheral circuit.

## 3. Partial Differential Equation

A partial differential equation is any equation with a function of multiple variables and their partial derivatives [23]. The function u: (1)$$u=u\;\left(t\;,\;x_{1}\;,\;\cdots ,\;x_{n}\right)$$
and the Partial differential equation can be defined as: (2)$$g\left(t\;,\;x_{1}\;,\;\cdots ,\;x_{n}\;,\;u\;,\;\frac{\partial u}{\partial t}\;,\;\frac{\partial u}{\partial x_{1}}\;,\;\cdots ,\;\frac{\partial u}{\partial x_{n}}\;,\frac{\partial^{2}u}{\partial t^{2}}\;,\;\cdots \right)=0$$

Typical results of partial differential equations have two forms: the analytical solution and the numerical solution. The analytical solution that can be expressed by an analytical expression is an exact combination of finite common operations. Given any independent variable, its dependent variable could be solved, so the analytical solution is also known as the closed-form solution. The numerical solution needs to be calculated iteratively from the boundary condition step-by-step [24], and it is the emphasis of the method in PDEs solver research. With the decrease of the step size, the numerical solution will be more accurate. There are multiple numerical methods, including the Euler method, Runge-Kutta, finite-difference [25], finite-element method [26] and finite-volume method [23]. 

### 3.1. Numerical Methods

#### 3.1.1. Finite-Difference Method

The principle of the finite-difference method (FDM) is understandable: converting the continuous problem to its corresponding discrete form and getting results within a finite number of calculations. The core mechanism of FDM is to approximate the partial derivatives at each point using its nearby values based on Taylor’s theorem. There are three basic steps in the finite-difference method:
(1)Regional discretization. According to the appropriate step size, the domain that needs to be calculated is divided into finite grids and using the function values on discrete grid points to approximate the continuous function values. (2)Transformation of partial differential equations. Using the difference coefficient to approximate the exact derivatives. (3)Solution of partial differential equations. Bringing the boundary conditions into the equation and repeating calculations to solve a large number of equations. 

The first-order finite-difference of g(x) of variable x can be defined as: (3)Δg(x)=g(x+Δx)−g(x)
where Δx is step size, or the so-called spacing between two grid points, and the first-order difference coefficient of g(x) of variable x can be defined as:(4)$$\frac{dg\left(x\right)}{dx}\approx \frac{\Delta g\left(x\right)}{\Delta x}=\frac{g\left(x+\Delta x\right)-g\left(x\right)}{\Delta x}$$

So that the forward difference coefficient, backward difference coefficient and central difference coefficient can be expressed as:(5)$$\frac{g\left(x+\Delta x\right)-g\left(x\right)}{\Delta x},\frac{g\left(x\right)-g\left(x-\Delta x\right)}{\Delta x},\frac{g\left(x+\Delta x\right)-g\left(x-\Delta x\right)}{2\Delta x}$$

The finite-difference method uses the difference coefficient to approximate the exact derivative. Similarly, the second-order difference coefficient can be expressed as:(6)$$\frac{d^{2}g\left(x\right)}{dx^{2}}\approx \frac{g\left(x+\Delta x\right)+g\left(x-\Delta x\right)-2g\left(x\right)}{\Delta x^{2}}$$

Taking a simple one-dimensional heat diffusion equation without a heat source as an example, the equation and boundary conditions are as follows: (7)$$\{\begin{array}{c} \frac{\partial u}{\partial t}\left(x,t\right)=\frac{\partial^{2}u}{\partial t^{2}}\left(x,t\right)+f\left(x,t\right)\\ u\left(x,0\right)=\varphi \left(x\right)\;,\;f\left(x,t\right)=0\\ u\left(a,t\right)=v1\;,u\left(b,t\right)=v2\\ a\leq x\leq b\;,0\leq t\leq T \end{array}$$
where u(x,t) is the temperature at grid point x at time t, *a*^2^ is the thermal diffusivity,
f(x,t) is the heat source and v1 and v2 are the constant boundary conditions at grid point a and *b*, respectively. 

Then, the solution domain is divided into finite grids in Figure 3. The step size in the t-axis direction is taken as Δt, and the step size in the x-axis direction is taken as Δx. Therefore,
(8)$$x_{i}=a+\left(i+1\right)\Delta x\;,\;\Delta x=\frac{b-a}{N}$$
(9)$$t_{i}=0+\left(j+1\right)\Delta t\;,\;\Delta x=\frac{T}{M}$$

According to (4) and (6), the continuous function (7) will change to the function of values on discrete grid points: (10)$$\frac{u\left(x_{i},t+\Delta t\right)-u\left(x_{i},t\right)}{\Delta x}=a^{2}\frac{u\left(x_{i}+\Delta x,t\right)+u\left(x_{i}-\Delta x,t\right)-2u\left(x_{i},t\right)}{\Delta x^{2}}$$
(11)$$u\left(x_{i},t+\Delta t\right)=u\left(x_{i},t\right)+\frac{\Delta t\cdot a^{2}}{\Delta x^{2}}\left[u\left(x_{i}+\Delta x,t\right)+u\left(x_{i}-\Delta x,t\right)-2u\left(x_{i},t\right)\right]$$

Excluding the boundary values, when t=Δt, the actual temperature of each point is
(12)$$\begin{array}{c} u\left(x_{1},\Delta t\right)=\varphi \left(a\right)+\frac{\Delta t\cdot a^{2}}{\Delta x^{2}}\left[u\left(x_{2},\Delta t\right)+u\left(x_{0},0\right)-2u\left(x_{1},0\right)\right]\\ u\left(x_{2},\Delta t\right)=\varphi \left(a+\Delta x\right)+\frac{\Delta t\cdot a^{2}}{\Delta x^{2}}\left[u\left(x_{3},\Delta t\right)+u\left(x_{1},0\right)-2u\left(x_{2},0\right)\right]\\ \vdots \\ u\left(x_{N-1},\Delta t\right)=\varphi \left(b\right)+\frac{\Delta t\cdot a^{2}}{\Delta x^{2}}\left[u\left(x_{N},\Delta t\right)+u\left(x_{N-2},0\right)-2u\left(x_{N-1},0\right)\right] \end{array}$$

The above system of linear equations contains N−1 equations, and we can rewrite them into a matrix equation: (13)$$\left[\begin{array}{c} u_{1}^{\Delta t}\\ u_{2}^{\Delta t}\\ \vdots \\ u_{N-1}^{\;\Delta t} \end{array}\right]=\left[\begin{array}{c} u_{1}^{0}\\ u_{2}^{0}\\ \vdots \\ u_{N-1}^{\;0} \end{array}\right]+\frac{\Delta t\cdot a^{2}}{\Delta x^{2}}\cdot \left[\begin{array}{cccc} 2 & -1 & \cdots & 0\\ -1 & 2 & -1 & \vdots \\ \vdots & -1 & 2 & -1\\ 0 & \cdots & -1 & 2 \end{array}\right]\left[\begin{array}{c} u_{1}^{0}\\ u_{2}^{0}\\ \vdots \\ u_{N-1}^{\;0} \end{array}\right]$$

(13) can also be expressed as: (14)$$U^{t+\Delta t}=U^{t}+AU^{t}$$
where A is the coefficient matrix, $U^{t}$ is the matrix at time t and
$U^{t+\Delta t}$ is the matrix at time t+Δt. 

The one-dimensional equation is the most ideal and simplest physical scenario. However, researchers and engineers need to solve the problems of two-dimensional or even multidimensional space aimed at the actual requirements. At this time, the two-point difference method is not suitable. As an example, there is a two-dimensional equation, a Laplace equation: (15)$$\{\begin{array}{c} \frac{\partial^{2}u}{\partial x^{2}}\left(x,y\right)+\frac{\partial^{2}u}{\partial y^{2}}\left(x,y\right)=0\\ u\left(a,y\right)=\phi_{1}\left(y\right)\;,u\left(b,y\right)=\phi_{2}\left(y\right)\\ u\left(x,c\right)=\phi_{3}\left(y\right)\;,u\left(x,d\right)=\phi_{4}\left(y\right)\\ a\leq x\leq b\;,c\leq y\leq d \end{array}$$

According to (6), the Laplace equation can be written as: (16)u(x+Δx,y)+u(x−Δx,y)+u(x,y+Δy)+u(x,y−Δy)−4u(x,y)=0
where the step size in the x-axis direction is taken as Δx, and the step size in the y-axis direction is taken as Δy. In Figure 4, the function value of each point can be calculated by the function values of its four neighboring points, and the whole domain is divided into N×N grid points. 

The equation in the grid point of (i,j) can be expressed as: (17)$$u_{i+1,j}+u_{i,j+1}-4u_{i,j}+u_{i-1,j}+u_{i,j-1}=0$$

Thus, (15) can be transformed into a matrix form as:(18)$${\left[\begin{array}{c} u_{-1,0}+u_{0,-1}\\ u_{-1,1}\\ u_{-1,2}+u_{0,3}\\ \vdots \\ u_{N-1,N}+u_{N,N-1} \end{array}\right]}_{N^{2}\times 1}={\left[\begin{array}{cccc} B & -I & \cdots & 0\\ -I & B & -I & \vdots \\ \vdots & -I & B & -I\\ 0 & \cdots & -I & B \end{array}\right]}_{N^{2}\times N^{2}}{\left[\begin{array}{c} u_{0,0}\\ u_{0,1}\\ u_{0,2}\\ \vdots \\ u_{N-1,N-1} \end{array}\right]}_{N^{2}\times 1}$$
where
(19)$$B={\left[\begin{array}{cccc} 4 & -1 & \cdots & 0\\ -1 & 4 & -1 & \vdots \\ \vdots & -1 & 4 & -1\\ 0 & \cdots & -1 & 4 \end{array}\right]}_{N\times N}\;,\;I={\left[\begin{array}{cccc} 1 & 0 & \cdots & 0\\ 0 & 1 & 0 & \vdots \\ \vdots & 0 & 1 & 0\\ 0 & \cdots & 0 & 1 \end{array}\right]}_{N\times N}$$

Consequently, (18) can also be expressed as: (20)$$U^{b}=AU$$
where A is the coefficient matrix, U is the matrix in grid point of (i,j) and $U^{b}$ is the matrix composed of the boundary conditions. 

#### 3.1.2. Runge-Kutta Method

The Runge-Kutta method includes two kinds of methods: second-order Runge-Kutta and fourth-order Runge-Kutta. 

Second-Order Runge-Kutta: $k_{1}$ and $k_{2}$ can be calculated by the following equations,
(21)$$k_{1}=h\times f\left(x_{n},y_{n}\right)$$
(22)$$k_{2}=h\times f\left(x_{n}+h,y_{n}+k_{1}\right)$$

The solution of the PDE can be expressed as: (23)$$y_{n+1}=y_{n}+\frac{1}{2}\left(k_{1}+k_{2}\right)$$

Fourth-Order Runge-Kutta: $k_{1}$, $k_{2}$, $k_{3}$ and $k_{4}$ can be calculated by the following equations,
(24)$$k_{1}=h\times f\left(x_{n},y_{n}\right)$$
(25)$$k_{2}=h\times f\left(x_{n}+\frac{h}{2},y_{n}+\frac{k_{1}}{2}\right)$$
(26)$$k_{3}=h\times f\left(x_{n}+\frac{h}{2},y_{n}+\frac{k_{2}}{2}\right)$$
(27)$$k_{4}=h\times f\left(x_{n}+h,y_{n}+k_{3}\right)$$

The solution of the PDE can be expressed as: (28)$$y_{n+1}=y_{n}+\frac{1}{6}\left(k_{1}+2k_{2}+2k_{3}+k_{4}\right)$$

### 3.2. Matrix Iterative Methods

After getting the matrix equations, the next main task is the iterative computation of a large-scale matrix, and common iterative methods include the Jacobi method, the Guass Seidel method and the SOR method. 

#### 3.2.1. Jacobi Method

The principle of the Jacobi method is to disassemble the coefficient matrix A into a diagonal matrix D, a negative upper triangular matrix U and a negative lower triangular matrix L. Consequently, A can be written as: (29)A=D−L−U
(30)$$D=\left[\begin{array}{cccc} a_{1,1} & 0 & \cdots & 0\\ 0 & a_{2,2} & 0 & \vdots \\ \vdots & 0 & \ddots & 0\\ 0 & \cdots & 0 & a_{n,n} \end{array}\right]\;,\;L=\left[\begin{array}{cccc} 0 & 0 & \cdots & 0\\ -a_{2,1} & 0 & 0 & \vdots \\ \vdots & \ddots & \ddots & 0\\ -a_{n,1} & \cdots & -a_{n,n-1} & 0 \end{array}\right]\;,\;U=\left[\begin{array}{cccc} 0 & -a_{1,2} & \cdots & -a_{1,n}\\ 0 & 0 & 0 & \vdots \\ \vdots & 0 & \ddots & -a_{n-1,n}\\ 0 & \cdots & 0 & 0 \end{array}\right]$$

For an equation of a matrix like B=A·X, replacing the coefficient matrix A, it will change to:(31)B=(D−L−U)·X
(32)$$X=D^{-1}\left(L+U\right)X+D^{-1}B$$

After the iterative calculation, the calculation result of the (K+1)th iteration is
(33)$$X^{\left(k+1\right)}=D^{-1}\left(L+U\right)X^{\left(k\right)}+D^{-1}B$$

#### 3.2.2. Guass Seidel Method

The principle of the Guass Seidel method is similar to the Jacobi method, where the difference is the derivation process, the Guass Seidel method rewrites B=A·X into: (34)$$X={\left(D-L\right)}^{-1}UX+{\left(D-L\right)}^{-1}B$$

After the iterative calculation, the calculation result of the (K+1)th iteration is
(35)$$X^{\left(k+1\right)}={\left(D-L\right)}^{-1}UX^{\left(k\right)}+{\left(D-L\right)}^{-1}B$$

In most cases, the Guass Seidel method converges faster than the Jacobi method, and only one set of storage units is needed to store ${\left(D-L\right)}^{-1}$, but $D^{-1}\left(L+U\right)$ and $D^{-1}$ are required to store in the Jacobi method. 

#### 3.2.3. SOR Method

Based on the Guass Seidel method, a convergence factor ω is added to the SOR method in order to improve the convergence speed. The calculation result of the (K+1)th iteration is
(36)$$X^{\left(k+1\right)}={\left(D-\omega L\right)}^{-1}\left[\left(1-\omega \right)D+\omega U\right]X^{\left(k\right)}+\omega {\left(D-\omega L\right)}^{-1}B$$

#### 3.2.4. Krylov Subspace Method

For an equation of a matrix like B=A·X, the result X can be expressed as $A^{-1}\cdot B$ directly. However, if the matrix A has a large size or is a sparse matrix, $A^{-1}$ will be very hard to solve. The principle of the Krylov subspace method is to approximate $A^{-1}\cdot B$.
(37)$$A^{-1}\cdot B\approx \overset{m-1}{\underset{i=0}{\sum }}\beta_{i}A^{i}B=\beta_{0}B+\beta_{1}AB+\beta_{2}A^{2}B+,\;\cdots ,+\beta_{m-1}A^{m-1}B$$
where $\beta_{0}\;,\;\beta_{1}\;,\;\beta_{2}\;,\;\cdots ,\beta_{m-1}$ are unknown coefficients, the step size m is related to the accuracy of the approximation and is less than the dimension of matrix A. 

After the discussion above, the number of iterations used by the Jacobi method, the Guass Seidel method and the SOR method is decreasing, which also means the improving of the calculation accuracy. Additionally, in terms of the hardware consumption and hardware implementation, the Guass Seidel method is better than the others. Moreover, the Krylov subspace method sacrifices accuracy to improve speed and is generally used large-scale matrixes. Relatively speaking, the SOR method is the most accurate and efficient method. However, the matrixes could not be inputted into the ReRAM arrays iteratively as weights. Calculating the matrixes directly will waste computing power in the multiplication of zero elements, because the matrixes used in numerical computations are generally sparse matrixes which include more than 60% of zero elements. 

### 3.3. Rearrangement and Split

Rearrangement and split are proposed to solve the multiplication of sparse matrixes whose majority of elements are zero elements. There are a number of studies focusing on cutting or splitting matrixes to improve computational efficiency, while many matrixes cannot be cut or split directly all the time. Therefore, rearrangement of sparse matrixes is needed to change the layout of matrix elements. In sparse matrix-vector multiplication (SpMV), the rearrangement matrix and the splitting matrix can replace sparse matrixes with dense matrix operations in many cases, which can greatly save memory and reduce computational overhead [27]. Moreover, the methods reducing the bandwidth of sparse matrixes in SpMV are quite useful for matrixes got from the PDEs. Taking a 1024×1024 sparse matrix as an example in Figure 5, the gray part of the matrix is composed of zero elements, and the blue part is composed of nonzero elements. Firstly, the 1024×1024 sparse matrix is rearranged to a diagonal aggregation matrix (not a diagonal matrix). In addition, it is rewritten as a combination of B and I. At last, the matrix is divided into four types of 16×16 slices (a, b, c, d).

According to the discussion of the matrix iterative methods in Section 3.2, the number of iterations used by the Jacobi method, the Guass Seidel method and the SOR method is decreasing. Though the Guass Seidel method has a lower hardware consumption and the SOR method has the highest computational efficiency, utilizing the Jacobi method with the splitting matrix can not only improve the accuracy but also cut down the number of iterations efficiently. Because the principal concern is which method has fewer zero elements, the Jacobi method is still the best choice for the CIM-based PDEs solver at present. 

## 4. CIM-Based Partial Differential Equation Solver

With the discussion of matrix iterative methods, the problem of the PDEs will be changed to the multiplication of the large-scale matrix. Therefore, the essence of the PDEs Solver is to achieve the high-performance multiplication. The CIM-based PDEs Solver could be composed with input drivers, shifters, adders, a computing array and DAC/ADCs. Matrix A will be stored in the array as weight, and matrix X will be entered into the array as input. Because of the limited size and precision of the array, the array maps a single column of the matrix to serval columns of an array. After the shifters and adders, the results of the serval columns will be collected and then be quantified in ADCs. The ADCs in the CIM PDEs solvers are usually SAR ADCs to balance the area, power and speed. Recently, CIM-based partial differential equation solvers can be based on different CIMs, including ReRAM, SRAM, flash memory and PCM. Each of them has advantages and their own suitable calculating methods and circuits.

### 4.1. ReRAM-Based Partial Differential Equation Solver

The coefficient matrix has been divided into several slices, and next they will be written as resistance values into the ReRAM array. Matrix X will be entered into the ReRAM array as input, and the slices can be used several times without replacement. That means the ReRAM array rarely needs to be written. The larger the size of the slices, the smaller the number of slices with the decrease of write times. On the contrary, the times of the iterative calculation will increase, and the computational efficiency will be lower. After the multiplication in the simulation domain, ADCs are required to convert the analog signals into digital signals. Then, after the digital signal processing, the result will be iterative, as will the input of the ReRAM array, and the final result can be got from the ReRAM-based partial differential equation solver. 

In the work of Mohammed A. Zidan, a general memristor-based partial differential equation solver is proposed with the finite-difference method. To solve the general matrix Equation (20), they use the Jacobi method to decrease the calculation of zero elements. Their memristor is composed of a Ta top electrode, a Pd bottom electrode and a thin Ta_2_O_5−x_ metal oxide. Additionally, the memristor crossbar has extremely high energy efficiency and area utilization, but a lower accuracy because of the variation. With the write-verify approach, they decrease the conductance variation from 5.3% to 0.85%, which overcomes the accuracy defects of the ReRAM immensely. They divide the matrix into equally sized slices, and practical crossbar sizes can be mapped onto the active slices exactly. Cutting the matrix not only minimizes the effects of the series resistance, sneak currents and virtual grounds, but also reduces the operation of zero elements [23]. With the memristor-based hardware and matching software system, they get high-precision computing results. 

Shichao Li and Wenchao Chen simulated fully coupled multiphysics based on bipolar resistive random-access memory in 2017 [28]. They utilized the finite-difference method and the Scharfetter-Gummel method to solve the PDEs, solving three fully coupled partial differential equations by the crossbar of the HfO_x_–based ReRAM [29]. Like the work of Mohammed A. Zidan, the accuracy has not been effectively improved, while more PDEs are discussed in this work.

In recent work by S. S. Ensan and S. Ghosh, a ReRAM-based linear first-order PDE solver (ReLOPE) is proposed to solve PDEs of the following form: (38)$$f\;\left(x,\;y\right)=y^{\prime }=ay+bx+c$$

Unlike the general memristor-based partial differential equation solver [23], ReLOPE is the first PDEs solver purely based on hardware and used only for linear first-order PDEs [20]. Moreover, the principle of the ReLOPE is based on the second-order Runge-Kutta method. Though theoretically fourth-order Runge-Kutta offers higher accuracy, the value of the resistance tremendously interferes with the iteration under the limitation of the accurate programming of the ReRAM. Substituting (3) and (4) into (37), (37) will change to: (39)$$y_{n+2}-y_{n+1}=\frac{2+2h\cdot a+h^{2}\cdot a}{2}\left(y_{n+1}-y_{n}\right)+\frac{2h\cdot b+h^{2}\cdot a\cdot b}{2}\left(x_{n+1}-x_{n}\right)$$

Similarly, substituting Equations (3)–(6) into (37), (37) will change to: (40)$$\begin{array}{c} y_{n+2}-y_{n+1}=\left(1+\frac{2}{3}h\cdot a+\frac{1}{3}h^{2}\cdot a^{2}+\frac{h^{3}\cdot a^{3}+2h^{3}\cdot a^{2}+h^{4}\cdot a^{4}}{24}\right)\left(y_{n+1}-y_{n}\right)\\ +\left(\frac{1}{6}b+\frac{1}{4}h\cdot b+\frac{1}{8}h^{2}\cdot a\cdot b+\frac{h^{3}\cdot a^{2}\cdot b+2h^{3}\cdot a^{2}\cdot b+h^{4}\cdot a^{3}\cdot b}{24}\right)\left(x_{n+1}-x_{n}\right) \end{array}$$

Figure 6 is the overview of ReLOPE. ReLOPE includes a fully ReRAM crossbar-based CIM, shifters, adders and DAC/ADCs. It expands the operating range of the solution by exploiting shifters to shift input data and output data. ReLOPE improves its power consumption, solving a PDE by 31.4×. The above methods with ReLOPE used have two limitations: (1) it is unachievable to program RRAMs with this accuracy (six decimal points for fourth-order Runge-Kutta in the paper of ReLOPE) at the current technical level; (2) the accuracy has loss due to the nonlinear variation of resistance with voltage and the iterative use of ADC. Therefore, ReLOPE cannot further improve its accuracy with the method of Runge-Kutta. 

### 4.2. SRAM-Based Partial Differential Equation Solver

Yannis Tsividis et al. proposed a programmable, clockless, continuous-time 8-bit hybrid (mixed analog/digital) architecture (ADC + SRAM + DAC) for solving ordinary and partial differential equations. The architecture, shown in Figure 7, is used to achieve nonlinear functions. The system consisted of an analog multiplier and analog adder/subtractor. The hybrid nonlinear function generator achieves 16× lower power dissipation and the computational accuracy of about 0.5% to 5%. 

In 2019, Thomas Chen and Jacob Botimer proposed a SRAM-based accelerator for solving PDEs [30]. They reformulated the multigrid Jacobi method in a residual form. By interleaving coarse-grid iterations with fine-grid iterations, their system reduced low-frequency errors to accelerate convergence. Their system contains 4 MAC–SRAMs, and each is a 320 × 64 8T SRAM array. The architecture is shown in Figure 8a,b. A year later, they updated the mapping of the MAC-SRAM [31] on the basis of their previous research. Finally, the SRAM-based accelerator achieved 56.9–GOPS, consuming 16.6 mW at 200–MHz. However, the SRAM-based CIM has the same accuracy problems due to limited multiplicand precision and limited ADC resolution. 

### 4.3. Flash Memory-Based Partial Differential Equation Solver

Jiezhi Chen et al. proposed a flash memory-based CIM hardware system to improve the computation efficiency of the time-dependent partial differential equations [32]. Based on the FDM and Jacobi algorithm, they got the matrix equation, then the coefficient matrix was mapped into the flash memory array as threshold voltages. Matrix X is transformed into pulse time as input. Compared with ReRAM, flash memory enables the realization of vector-matrix multiplication with high accuracy and a good tolerance for device error. Moreover, it also has the advantages of ultra-high density and low cost. 

### 4.4. PCM-Based Partial Differential Equation Solver

In 2018, Manuel Le Gallo and Abu Sebastian et al. used mixed-precision in-memory computing which combined a von Neumann machine with a computational memory unit to solve PDEs [33]. The mixed-precision CIM PDEs solver uses a low-precision computational memory unit to obtain the approximate solution of the first part and high-precision processing iteratively to improve accuracy in the second part. It achieves the low-precision matrix-vector multiplication by using a PCM crossbar array based on the iterative Krylov-subspace method. The PCM-based PDEs solver could offer up to 80 times lower energy consumption than the FPGA solution because of the architecture of PCM-based CIM and the mixed-precision system. 

### 4.5. Discussion of Partial Differential Equation Solver

The memristive crossbar CIM PDEs solver based on the PCM chip could already offer up to 80 times lower energy consumption than the FPGA solution. The energy efficiency and speed of the memristive crossbar CIM PDEs solver is several hundred times than the PDEs solvers based on IBM POWER8 central processing unit (CPU) and NVIDIA Titan RTX graphics processing unit (GPU) [26,34]. However, the CIM-based PDEs usually solves the 4-bit to 8-bit PDEs computation in satisfaction of accuracy requirements.

Table 2 summaries the representative CIM-based PDEs solvers recently and their performance are compared. 

The deficiencies of ReRAM, including low inherent accuracy, nonlinearity and susceptibility to environmental changes, directly determine the use of ReRAM for high-precision computing. With the constant development of process levels, ReRAM devices will be manufactured accurately and get close to high-precision computing to a certain extent. At the same time, the write-verify method or multiread/write method can also solve accuracy problems, but with an increase in latency. Both of them have a lot of worth in research in ReRAM-based PDEs solvers in the future. To reduce the memristor device variability and nonlinearity, in the work of Mohammed A. Zidan, a general memristor-based partial differential equation solver used a write–verify method to write and update the coefficient values in the crossbar. The write-verify operation is based on a sequence of write-read pulse pairs, each pair including a programming (set or reset) pulse and a subsequent read pulse. When the conductance reaches within a predetermined range of the target value, the write operation is considered complete. The write-verify feedback method could decrease the cell-to-cell variation of <1% from 5.3%. 

Though flash memory and SRAM have higher accuracy and density compared with ReRAM for CIM in numerical computations, ReRAM with nonvolatile and multivalued characteristics is still the most extensively studied for CIM-based PDEs solvers. 

The sparse coefficient matrix, with only a small number of nonzero elements, usually has a large matrix size. The coefficient matrix must be divided into a certain number of slices, whose sizes are typically 16×16 or 32×32. When the ReRAM array, having a huge size and multiple rows or columns, are unopened, the result of the ReRAM array may have a large error because of the unopened leakage currents. Hence, the size of the ReRAM array, and the size of the slices above-discussed, not only depends on the system working, but also is determined by the hardware operation of the ReRAM array. Moreover, the collected currents need ADC for the next computation, and there are losses of accuracy in the process. Other CIM-based PDEs solvers also have similar problems. 

How to cut apart the matrix into slices, how small the size of crossbar arrays is and how the single array is to work are urgent problems to be solved at the system level. At the same time, rearrangement of sparse matrixes and SpMV will play an important role in the future of CIMs for high-precision computing tasks. 

## 5. Summary and Outlook

Different from neural network computations, numerical computations are used in early, basic disciplines, receiving general attention from researchers all the time. In the past few years, the research has become more and more popular to solve numerical computations using CIM. CIM for numerical computations improves energy efficiency and computational efficiency. All kinds of CIM PDEs based ReRAM, SRAM, flash memory and PCM have been proposed with various characteristics. The recent developments of CIM for numerical computations were compared. This article described the ReRAM-based CIM technology in detail. Then, it reviewed the numerical methods of PDEs and matrix iterative methods. Finally, the future of CIM for numerical computations can be summarized as follows: Regardless of which array of CIM, the accuracy is still the biggest challenge;More research of CIM-based numerical computations should focus on the computational methods of sparse matrixes;As for matrix iterative methods, the principal concern is which method has fewer zero elements, so the Jacobi method is still the best choice for CIM-based PDEs solvers at present. In addition, the Krylov subspace method is better when solving very large-scale matrixes;The future of CIM for high-precision computing tasks really needs a software/hardware codesign to collaborate the algorithm and the CIM array.

## Figures and Tables

**Figure 1 micromachines-13-00731-f001:**
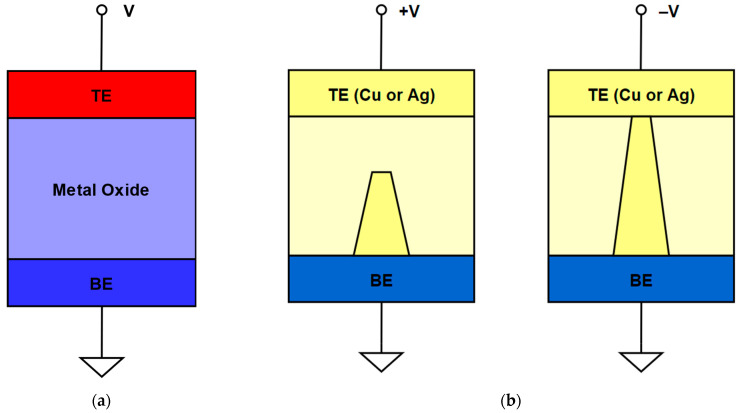
(**a**) Structure of OxRAM; (**b**) Structure of CBRAM.

**Figure 2 micromachines-13-00731-f002:**
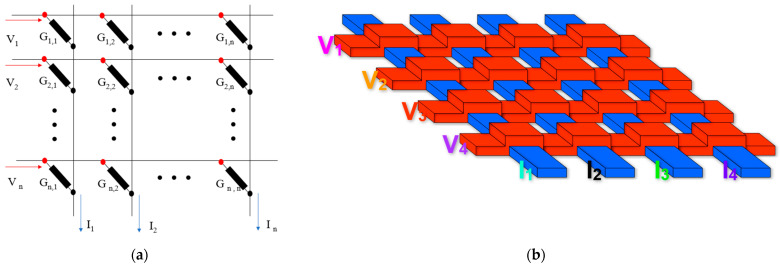
(**a**) Schematic diagram of ReRAM crossbar; (**b**) Structure of ReRAM crossbar.

**Figure 3 micromachines-13-00731-f003:**
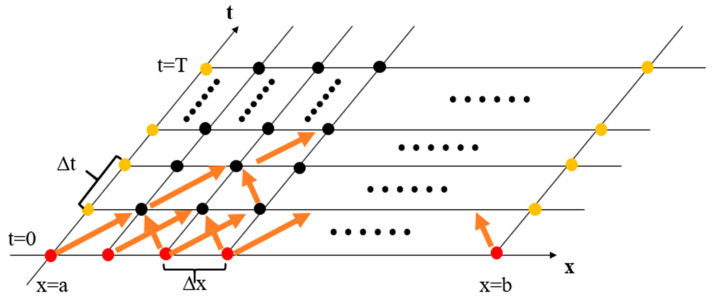
Lattice graph of a one-dimensional heat diffusion equation.

**Figure 4 micromachines-13-00731-f004:**
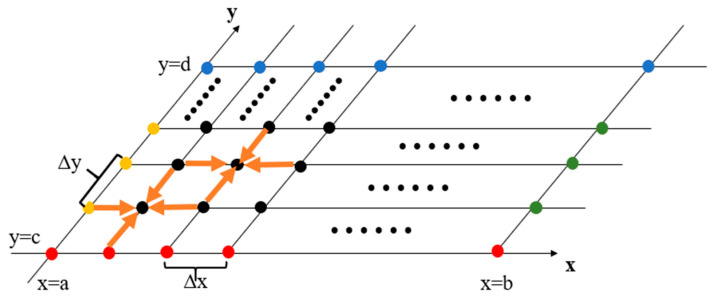
Lattice graph of a Laplace equation.

**Figure 5 micromachines-13-00731-f005:**
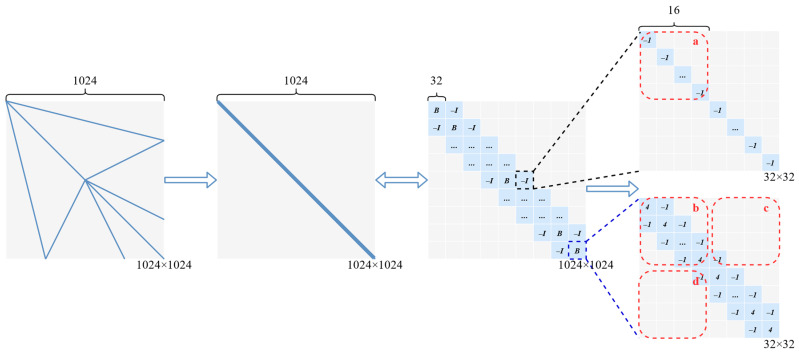
The process that a 1024×1024 sparse matrix is divided into slices.

**Figure 6 micromachines-13-00731-f006:**
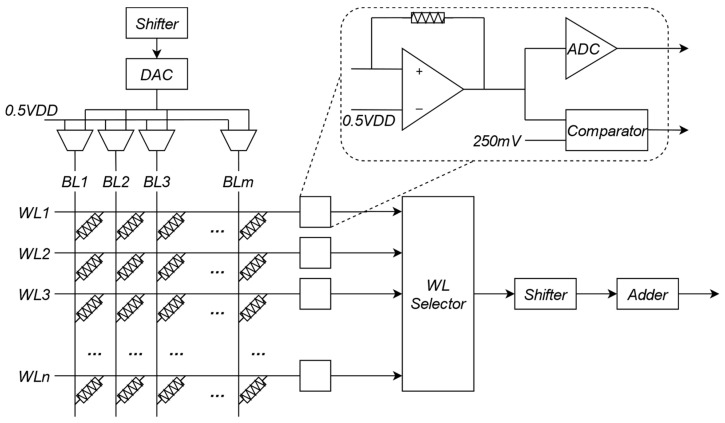
Overview of ReLOPE.

**Figure 7 micromachines-13-00731-f007:**
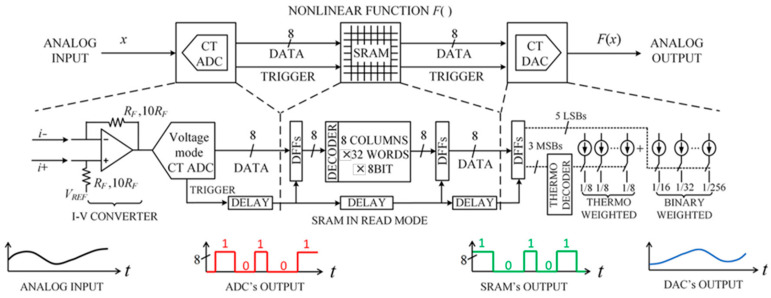
The continuous–time programmable nonlinear function generator.

**Figure 8 micromachines-13-00731-f008:**
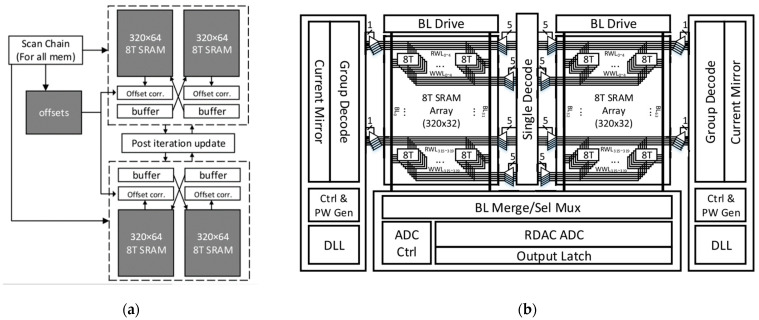
(**a**) Architectural sketch of PDE iteration module; (**b**) Block diagram of MAC SRAM.

**Table 1 micromachines-13-00731-t001:** The types of the NVM and the category of ReRAM.

ReRAM	MRAM	FeRAM	PCM	Flash Memory
OxRAM	STT-MRAM SOT-MRAM VCMA	FTJ		NAND Flash Nor Flash AG-AND Flash
CBRAM				

**Table 2 micromachines-13-00731-t002:** Comparation of several representative CIM-based PDEs solvers and GPU.

Type of CIM	Reference	Technology Node	Energy Efficiency	Accuracy	Latency
ReRAM	Sina Sayyah Ensan 2021 VLSI	65 nm	31.4×	11-bit (97%)	25 ns
	Mohammed A. Zdan 2018 NE	NA	NA	64-bit	1 us
SRAM	Thomas Chen 2020 JSSC	180 nm	0.875 TOPS/W	32-bit	90 ns
	Ning Guo 2016 JSSC	65 nm	16×	18-bit (95%) 8-bit (99.5%)	NA
NOR Flash Memory	Yang Feng 2020 SNW	65 nm	NA	64-bit	NA
PCM	Manuel Le Gallo 2018 NE	90 nm	24×	mixed-precision	<100 ns
GPU	NVidia Titan RTX	12 nm FFN	0.06 TOPS/W	64-bit	NA

## Data Availability

Not applicable.

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
