# Peer review of "Compute-in-Memory for Numerical Computations"

_micromachines, 2022, doi:10.3390/mi13050731_

Round 1
Reviewer 1 Report
The authors try to review use of RRAM CIM for PDE. While the topic is relevant and important. The paper needs significant rework:
- The paper contains many grammatically incorrect statements like -the 3-D vertical 92 ReRAM was emerged in 2009. There are many such statements.
- The paper while provides good overview of the math of PDEs, but fails to critically analyze different RRAM works. The RRAM PDE works are mentioned as if they are the abstract from respective paper, without any mention if drawback, challenges, novelty etc
- The paper also does not do justification to non-idealities of RRAM, and outlook on where such PDEs can be used. How non-idealities can be mitigated, what is the mitigation cost etc
Reviewer 2 Report
The authors present a review about computing in memory (CIM) for solving partial differential equations (PDE). Such review is useful given the recent developments in this field, but following issues should be addressed.
- The manuscript introduces the numerical methods, and then some examples of using CIM for PDE solver. But the link between them is missing. It is suggested to provide more details about how CIM array can be adapted to do the numerical calculation, such as weight mapping, sensing techniques, etc., before introducing the examples.
- The CIM based PDE solver should be compared with non-CIM computing techniques, to show the difference in energy efficiency, accuracy, latency, as in Table 2.
- It is suggested to introduce the potential application scenarios for using CIM based numerical computation.
Round 2
Reviewer 1 Report
The authors have taken care of my concerns.